# Prioritizing Work Health, Safety, and Wellbeing in Corporate Strategies: An Indicative Framework

**Brent Halliday** [1,*] **, Luke van der Laan** [2] **and Aldo Raineri** [3]

1   Faculty of Business, Education, Law and Arts, University of Southern Queensland, Toowoomba Campus, Toowoomba, QLD 4350, Australia
2   School of Humanities and Communications, University of Southern Queensland, Toowoomba Campus, Toowoomba, QLD 4350, Australia; luke.vanderlaan@usq.edu.au
3   School of Health, Medical and Applied Sciences, Central Queensland University, Brisbane Campus, Brisbane, QLD 4001, Australia; a.raineri@cqu.edu.au
*   Correspondence: brent@upsidesafety.com.au

**Abstract:** As a prominent organizational issue, there was limited evidence in the literature regarding the relationship between organizational strategy, workplace health, safety, and wellbeing, and performance measurements that demonstrate a measurable impact on organizational performances. Based on this gap in the literature, the purpose of the study was to examine business practices, health, safety, and wellbeing practices, and measurement systems to inform the development of a health, safety, and wellbeing strategy and employee engagement framework in order to add strategic value to businesses beyond standard practice. An exploratory mixed methods study, consisting of eight semi structured interviews and ninety-five survey responses from a cross section of private and public sectors leaders and health, safety, and wellbeing and human capital professionals was undertaken. Thematic analyses and exploratory factor analyses revealed a seven-factor health, safety, and wellbeing strategy framework that integrates key concepts, resilience engineering, wellbeing, health and safety management, employee engagement, risk management, and corporate governance. The final strategy framework provides empirical evidence supporting a suitable framework for businesses to improve individual and organizational performance.

**Keywords:** work health and safety; wellbeing; strategy; engagement; resilience

## 1. Introduction

The safety and well-being of an organization's employees are key to achieving the goals of its strategy and the broader organizational performance [1]. As such, organizations have a role in ensuring that employees are in an optimal state of well-being and engagement, including having decent meaningful work [2,3] that enables them to realize strategic goals. Organizations increasingly recognize the importance of the relationship between employee safety and wellbeing and strategy success, which ultimately benefits both the individual and organization.

However, traditional indicators suggest that the human and financial costs of work-related injury, illness, chronic disease, and poor mental health are significant detractors of optimal performance in Australia [4,5]. Workplace health, safety, and wellbeing (HSW) are not limited to individual consequences only, but can also affect the financial viability and investment, the wider community in which business operates, or the organizational reputation [6].

As such, HSW is a strategic imperative that organizations must respond to, including being on the "corporate agenda" as a governance and strategic priority [7]. The recognition of the broader impact of HSW also fits with concepts such as corporate social responsibility [6,8], license to operate [6], and legitimacy obligations, which also represent a shift away from the focus on risk and risk mitigation as it relates to workplace health and safety.

HSW has largely remained detached from organizational strategy. Organizational-level HSW is a challenging area due to the operational focus of corporate strategies and plans which are aimed at improving workplace-level performance and productivity. There is limited evidence reported in the extant literature connecting HSW to organizational strategy and its potential contribution to achieving organizational goals. Yet, achieving business goals is increasingly and empirically associated with the subsequent performance [9]. As such, there is a clear need to conceptualize the relationship between enhancing the HSW capacity and strategic performance in achieving organizational goals.

Surprisingly, organizations have had little insight into how HSW adds value in achieving organizational strategic goals, as much of the traditional focus of workplace health and safety has been dominated by a risk management and mitigation approach. The dominant aim has been to minimize the potential for unexpected negative outcomes in the workplace such as avoiding the costs associated with workplace injuries, disability, and illness. Consequently, the potential added value of HSW as a strategic priority has resulted in numerous commentators reporting on the "benefits" of HSW from both the operational and strategic perspectives [10–12]. What is less known, representing a gap in the extant literature, is a practice perspective as to how HSW can be strategically operationalized in order to achieve optimal organizational outcomes when these outcomes have traditionally been framed in terms of the "benefits" derived from the prominent risk management approach to HSW.

## 1.1. The Benefits of HSW

HSW definitions reported in the literature often vary according to disciplinary perspectives. For example, safety has been defined as 'The application of hazard control through the workplace, person and system by integrating into the organization sustained actions, accountability and reducing risk to as low as reasonably practicable to mitigate potential injury' [13] (p. 68) and the '…ability for a system to perform its intended purpose, whilst preventing harm to persons' [14] (p. 1), whereas health can be defined as 'a state of complete physical, mental, and social well-being and not merely the absence of disease or infirmity' [15]. And, in the broader context, work wellbeing has been defined as '…all aspects of working life, from the quality and safety of the physical environment, to how workers feel about their work, their working environment, the climate at work and work organization' [16].

This mix of perspectives, it is argued, limits the development of HSW theory and practice and also the demonstration of the organizational benefits. Instead, an interdisciplinary approach which integrates the advancements made in these and other associated fields, such as wellbeing (both eudaimonic and hedonic perspectives), organizational risk and resilience, and human capital management, is needed.

Accordingly, for the purpose of this study, and in seeking to promote an interdisciplinary approach, HSW is defined as 'a state facilitated by the organization which enables good work-related social connection, promotes physical and psychological health, satisfaction with the job, and personal growth. This leads to optimal worker motivation and engagement resulting in positive outcomes in an employees' working life, social life, and organizational performance' [17].

The current notion of "benefits" has largely employed a broad cross-section of reporting indicators such as reduced injuries and illness, reduced worker compensation costs, and improved productivity and compliance [18,19]. They have been predominantly associated with the aim of mitigating risk to achieve "zero accidents". More recently, the focus has shifted to areas such as safety management systems, leadership and culture, and performance measurement in order to articulate the value of workplace health and safety [12,18]. However, such "benefits" appear to be limited in their ability to demonstrate the business value of HSW from a strategic perspective. This is especially apparent when considering that the performance improvement which is attributed to traditional workplace health and safety practices has plateaued in the quest for "zero accidents", aimed at reducing costs and improving productivity. From a practice perspective, it has become clear that the reliance

on traditional approaches, including hazard management, safety management systems, human factors (physical and psychological attributes and considerations), and culture is not going to achieve the desired level of health and safety performance improvement alone [20], or contribute to the strategy outcomes of an organization. Hence, greater governance and oversight and more resilient business controls [21,22], including a reconceptualization of HSW measurements that focus on the presence of capacity and positive work outcomes [14], have shifted to becoming a strategic imperative.

In recognition of the limitations of the traditional approaches (i.e., risk mitigation), views such as high-reliability organizations [23], resilience engineering [24], and "Safety 2" [25] have emerged to illustrate new thinking regarding organizational safety and its management. In contrast to the traditional focus of HSW and its benefits, these views include a focus on aspects such as organizational capacity, adaptable risk models that enable the use of resources (including employees) in a proactive manner to balance safety and operational aspirations [14], examining how work is undertaken under normal conditions, and proactively learning from both failure (e.g., accidents) and success (e.g., high risk work completed without accidents). Notably, these approaches fit with the notion that HSW has moved to the "resilience age", where flexible robust approaches enable adjustment to work, conditions and system variations, and changes [26].

Although being extremely valuable in changing the focus, these frameworks appear to remain safety-centric with health and wellbeing being cast as prominent occupational and societal risks [27–29], rather than a strategic opportunity for organizations to realize their aspirations.

### 1.2. Strategy and HSW

It is evident that new holistic business strategy-centric approaches are required [30,31] that integrate HSW and enhance engagement and discretionary efforts to leverage the talent of employees in order to engage with organizational strategic priorities and improve organizational outcomes. These approaches need not only mitigate pathologies and risk but support positive individual and organizational outcomes through a shift to capacity and wellness and wellbeing thinking [32].

From a human capital perspective, there is well-established evidence that employee health and well-being, engagement, and individual and organizational performance outcomes are systemically linked [3,32]. However, high levels of work and job dissatisfaction continue to be reported, suggesting that increasing disengagement is a significant human capital issue for business [33]. Coupled with declining levels of physical and psychological health, this has resulted in employees seeking a more balanced working life that supports greater overall well-being. When employees perceive that they are unable to achieve a greater work/life balance, they default to disengaging from their work to accommodate a greater sense of well-being. This has significant implications within the context of realizing organizational aspirations. Central to addressing this common issue is having a safe working environment and conditions [3,34], a sense of work-related personal growth and accomplishment, and engagement that is derived from decent meaningful work [2,3,35].

From an organizational strategy perspective, the need to maintain high levels of employee engagement to realize organizational aspirations resonates with the commonly held assumption that an organization's employees are critical to strategy development. This is because engagement is associated with the generative and synergistic value that motivated employees contribute to organizational systems [36]. Not only does higher engagement contribute to anticipating the future and the creation of enabling work environments [14], but it is fundamentally necessary for innovation and the execution of organizational strategy [37]. Furthermore, engaged employees fulfil the role as the central actors within the HSW strategy implementation process [38].

To operationalize HSW as an enabler of strategy and innovation through engagement, employees must understand how their role contributes to the broader purpose of the organization. This is achieved by providing a "Line of Sight", defined 'as an employee's un-

derstanding of the organizations strategic goals and what actions are necessary' [39] (p. 500) in its execution. In contrast, a lack of meaning and purpose, role clarity, and alignment with organizational goals has been found to have a negative impact on personal wellbeing, engagement, and, ultimately, the achievement of organizational goals [40–42].

With such challenges in the operational environment, the authors suggest that there is (a) a strategic organizational performance incentive that is associated with increasing employee engagement within HSW, and (b) a clear need to shift thinking and practice, which is enabled by frameworks that place an emphasis on health and wellbeing within the organizations strategy in order to (c) achieve optimal individual outcomes that benefit the organization as described by the "mutual gains" hypothesis [43].

Despite the recognition that traditional HSW approaches are limited in their ability to realize strategic goals, to date there has been a paucity of empirical evidence about the relationship between organizational strategy, HSW, and organizational performance measurement. Models and frameworks that have attempted to illustrate the complexity of this proposed relationship are also rare. Whilst the literature reports numerous studies concerning the relationship between employee engagement and performance, they have not defined and operationalized HSW strategy within this relationship. This gap led to calls for further research into HSW from a business perspective [29,30,44].

How, then, can organizations strategically incorporate HSW imperatives to achieve improved performance outcomes within the context of the exponentially increasing prevalence of chronic disease, mental health issues, and job dissatisfaction?

A key premise of this paper is that a practice perspective for operationalizing HSW in an organization's corporate strategy is missing from the literature and necessary to gain traction in deriving strategic value for organizations. Drawing on prominent theories, models, and frameworks from the health and safety, wellbeing, human resources, and business management literature, this paper presents a HSW strategy and employee engagement framework that seeks to address this gap in the literature. It does so by integrating organizational and HSW strategy, employee engagement, and well-being in an attempt to optimize strategy realization. The purpose of this paper is twofold: firstly, as part of a broader mixed method study, it provides a brief review of the qualitative phase one of the study, and secondly, to report on the second quantitative phase that sought to determine the validity and reliability of the proposed HSW strategy and employee engagement framework within the context of an organizational strategy.

## 2. Literature Review

The objectives of the present study were to (a) establish operational definitions for HSW strategy and employee engagement in order to address the gaps in the existing literature; (b) to investigate the relationship between HSW strategy and employee engagement; and (c) to develop an industry-confirmed HSW strategy and employee engagement framework for the integration into organizational strategy within the Australian context. To frame the study, a further review of the literature based on the research question (RQ) 'RQ1: What business practices, HSW practices and measurement systems are suitable to inform the development of the HSW strategy and employee engagement framework to add strategic value to businesses?' was needed.

A critical analysis of a select literature was conducted. To ensure relevance to the study, a key word search of databases and online journals such as PsycInfo, Ebsco Google Scholar, Web of Science, JSTOR, Emerald, and Sage was completed using workplace health and safety, safety culture, wellness, wellbeing, strategy, strategic management, corporate governance, employee engagement, leadership, motivation, and safety and/or health and/or wellbeing performance measurement was completed. This led to a refined list of existing papers, journal articles, books, and conference proceedings, in addition to gray literature documents. A synthesis of the resultant literature was undertaken by reviewing each article with a focus on those that outlined a strategic HSW or employee engagement approach; discussed the causal or hypothesized relationships between employee wellbeing,

engagement, and business performance; discussed or reported on business approaches, such as corporate governance for HSW or employee engagement; or illustrated a shift in thinking from a risk mitigation focus to a health and wellbeing strategy.

The qualitative content analysis of the literature resulted in key theories, operational definitions, variables, and all potential attributes being established for each of the key constructs that emerged for this study. These were grouped and recorded in a Microsoft Word table, under the heading's key concepts; sub-variable; definitional themes; and attributes. The outcome of the literature review was an initial conceptual framework for testing through semi-structured interviews. Several models or frameworks pertaining to HSW approaches, in addition to well-developed theories for organizational strategy and employee engagement, were revealed. These are briefly highlighted, as they were deemed suitable to inform the objectives of the current study.

### 2.1. Organizational Strategy

The discussion on strategy formulation in the literature suggests that it is a complex process, comprising cognitive, behavioral, social, and technical dimensions [45]. A prominent theory of strategy formulation extensively reported in the literature associated with HSW is the resource-based view (RBV), with significant contributions coming from the work of Barney [46], Prahalad and Hamel [47], and recently Miller [48]. RBV adopts an "inside-out" view by focusing on the internal capabilities of the organization, tangible and intangible resources, and the core competencies needed to be competitive and sustainable [49]. The strengths of RBV include its ability to provide the initial direction for the organization and defines the resources required as "inputs" that enable an organization to carry out its activities, which are the primary source of a business profit [49]. Employee and the attendant organizational capabilities are considered a source of competitive advantage and are therefore central to the execution of the organization's strategy and the achievement of the strategic goals [50].

### 2.2. Workplace Safety

Safety science has evolved significantly through recent theories such as "Safety 2" [25] and resilience engineering [14], both of which have contributed to the reconceptualization of organizational safety management. As noted, these approaches support a more adaptive and responsive process, recognizing employees as being a crucial "solution" in maintaining the balance between organizational performance and safety. This is a far more positive approach than the traditional risk mitigation focus. It also aligns with the broader organizational resilience capability framework consisting of anticipation, coping, and adaptation phases outlined by Duchek [51]. This approach informed recent research by Klockner and Merdith [26] to measure an organization's resilience (safety) potential. From a strategic perspective, Carden et al. [52] outlined a model drawing on enterprise-wide risk management principles, suggesting that it is 'imperative for companies to manage unforeseen events and safety risks' (p. 143). The model is based on the input (safety risks)–process (corporate governance)–output (fewer incidents) quality management approach. Significantly, the approach was consistent with the wellbeing framework applied by Danna and Griffin [53].

More recently, Zou and Sunindijo [54] outline a strategic safety management approach applied to the construction and engineering sector that aligns with traditional business centric methods. Strategy formulation, they suggest, consists of phases related to (a) the circumstances in which strategy emerges (Strategy context), (b) how strategy content is developed (Strategy process), and (c) the dimensions, content, and outcome of the strategy process for implementation (Strategy content). Consistent with traditional business thinking, the framework includes a safety vision, goals, and core competencies, with the starting point for strategy development being an assessment of the strengths and weaknesses of existing safety strategies and the external environment to formulate new strategies and develop implementation and evaluation plans.

Similarly, the concept of "guided adaptability", informed by resilience engineering theory, is a shift from traditional thinking to a more proactive business centric approach, adopted to balance safety and organizational performance requirements through organizational capacity. Guided adaptability dimensions of anticipation, readiness to respond, synchronization, and proactive learning [14] support the creation, revision, and refinement of risk models in strategy development, meeting operational demands whilst maintaining safety. Such phases are notably consistent with organizational resilience, which has experienced increased interest from practice and academic perspectives in relation to organizational safety management [26].

### 2.3. Workplace Health and Wellbeing

Workplace health and wellbeing has evolved more recently as a priority focus for many businesses on the back of the COVID-19 pandemic, as well as issues such as the increasing prevalence of mental illness and aging workforces. Accordingly, both workplace health and safety and human capital disciplines have called for further research and practice-based health and wellbeing solutions (e.g., Shuck et al. [3]). Despite limited evidence, one strategic approach that focuses on the macro level of the enterprise is the Healthy Workplace Framework [55]. This framework establishes four avenues to address and promote holistic worker health, safety, and wellbeing, namely '(i) the psychosocial work environment (ii) the physical work environment (iii) personal health resources, and (iv) linkages between the enterprise and its wider community' (p. 83). Joss et al. [56] applied this framework in several Australian workplaces and found that there were positive associations with such interventions. Similar frameworks to follow this methodology include the Work Well Model [27] and National Institute for Occupational Safety and Health (NIOSH) Total Worker Health Model [57]. More recently, Shuck et al. [3] drew on the "social determinants of health" concept from the public health literature in order to introduce the "work determinants of health" term into human capital scholarship and practice. In doing so, a framework comprising stress, capacity, social and physical environment, and meaning and purpose was proposed to promote worker health and wellbeing. Notably, these dimensions align with the Healthy Workplace Framework outlined above. Furthermore, the inclusion of stress, whilst not new, fits with the recognition that employee recovery is a necessary component in promoting work-related wellbeing, strategically. One organization-centric theory that supports this is the job-demand resources (JDR) model [41], which seeks to optimize a balance between personal "inputs and outputs". JDR considers that the individual's resources such as job autonomy, supervisor support, and goal clarity create motivation, but once the demands of the role exceed the individual's ability to cope, their performance is reduced due to physical and psychological health impairment.

### 2.4. HSW Strategy Framework

Based on the review and critical analysis of the literature, an initial framework was developed to operationalize HSW within the organizational strategy context.

The preceding section established that the recent discussion in the literature indicates the nature of work has changed, and workplace health and safety practices need to enable both individual and organizational health, in addition to anticipating, adapting, and responding to risks in order to achieve the desired business outcomes. A shift is required through new approaches that connect HSW with the business through multidisciplinary solutions. This change also necessarily recognizes that people are crucial to business success, rather than being a risk to overcome, especially in strategy execution. As such, employees not only need to be in an optimal state that is free from injury and illness and in good mental health, but also thrive and are satisfied with their quality of work life, inclusive of positive rewarding work relationships (social connection) and decent work.

The initial framework design for the study incorporated perspectives from the HSW, business and human capital disciplines, and the relevant literature. The following con-

structs and operational definitions were established from the existing literature, ensuring the proper delineation of research based on the theory and the extant literature (Table 1).

**Table 1.** HSW strategy framework constructs and definitions.

| Construct | Operational Definition |
|---|---|
| Organizational Context | The set of organizational circumstances, under which the strategy process and content is determined to set the direction and scope of an organization over the long term. It is informed by how employees perceive the enactment of organizational policies and procedures relating to HSW in their organization at a given point in time and the organizational obligations beyond legal compliance [17]. |
| HSW Strategy | A strategic direction and allocation of resources dedicated to aligning internal capabilities with opportunities and threats to achieve a future state of HSW, as integrated into, and recognized as a priority of the organizational strategy, while being supported by the organizational mission, values, and priorities [17]. |
| HSW Employee engagement | A workplace approach designed to ensure that employees are committed to their organization's goals and values, motivated to contribute to organizational success, and are able, at the same time, to enhance their own sense of well-being through a positive, fulfilling, work-related state of mind that is characterized by vigor, dedication, and absorption [17]. |
| HSW Strategy efficacy | The combination of pre-defined results or other units of information which reflects, directly or indirectly, the extent to which an anticipated HSW outcome is achieved, or the quality of processes leading to that outcome. These may be (i) qualitative, which are indicators that would describe or assess a quality, or (ii) quantitative, which are an indicator that can be counted or measured and described numerically [58]. |
| Leadership | Strategic leadership is the ability to anticipate, envision, and maintain flexibility, think strategically, and work with others to initiate changes that will create a viable future for the organization [59]. |

Having verified the gap in the literature and derived the initial framework from the literature, it was necessary to pose RQ1 to respondents in order to determine the validity of the initial framework. A further research question, '(RQ2) How can the developed HSW strategy and employee engagement framework improve HSW and business performance beyond standard HSW management processes in businesses?' was posed, aimed at gaining further practice insights to confirm the revised frameworks suitability.

## 3. Methodology

The study adopted a pragmatic paradigm, in that it was problem focused and did not ascribe a particular ontology to its perspective. Rather, it sought to explore the nature of the problem (operationalizing HSW as part of organizational strategy) and provide evidence-based practice insight as to a possible response to the problem. The study adopted a sequential mixed methods approach, incorporating both qualitative (to gain a deeper understanding) and quantitative enquiry to validate the qualitative insights and triangulate the data.

The methodology is illustrated in Figure 1 below.

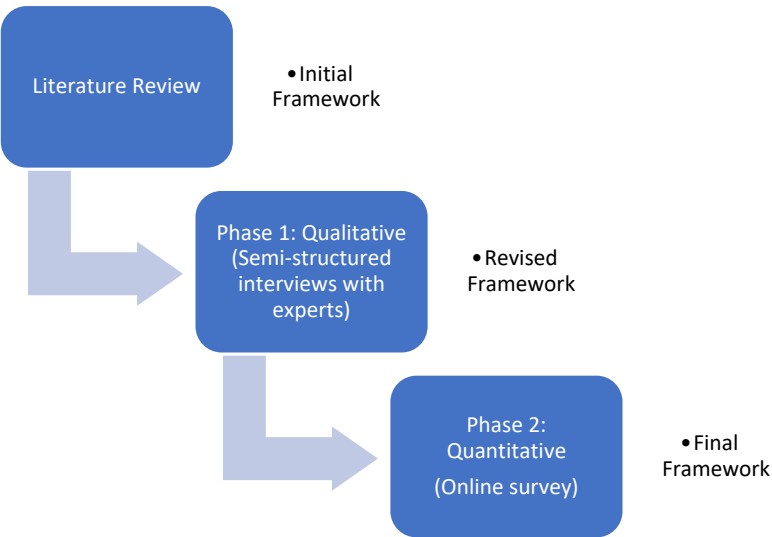

**Figure 1.** Research methodology.

*3.1. Summary of Phase One Semi Structued Interviews (Qualitaitve Enquiry)*

The objective of phase one was to develop a framework for HSW and employee engagement that has been deemed acceptable by the industry, based on the initial framework developed from the literature review.

The criteria used to select appropriate cases and delimit the sample group was based on participants having (1) current experience at a management level (middle or executive), in which they were responsible for the WHS workplace health and safety or wellbeing strategy, or (2) current experience at a management level (middle or executive), in which they were responsible for employee engagement or human capital management, and finally that they are (3) currently working in either public or private sector organizations.

Upon the notification of the ethical approval, an email invitation was sent to ten prospective participants from the first author's professional network with the consent forms, participant information sheets, and an overview of the research topic. To enhance the likelihood of participation, subject matter experts had the discretion of nominating the time and location for the interview, given these have been identified as potential barriers in conducting such research [60].

The semi-structured interview process was administered by the first author through a combination of face-to-face and telephone interviews, with the former being preferred, given its ability to elicit a higher participation rate. In total, eight interviews were conducted with each participant for a time of between 30–60 min. The male participants were from Queensland resources (1), manufacturing (1), public sector (5) industries and the South Australian (1) resources industry. Consistent with the guidance provided by Rowley [60], six key questions/themes were used to guide each interview (Appendix A). To ensure anonymity and confidentiality, information collected during the interviews was de-identified and allocated a unique code for transcription and further analysis. All interview records were maintained by the first author only.

On the completion of the interview phase, the data analysis commenced, with each audio interview being carefully transcribed verbatim and recorded in an excel spreadsheet database. The excel database used was structured in accordance with the profile matrix outlined by Kuckartz [61]. The transcribed data were then checked for accuracy against the recording prior to proceeding with manual analysis. The vertical columns represented each interviewee (1–8) and the horizontal categories represented each of the key constructs covered in the interview. The interview data were classified against worker wellbeing, organizational context, HSW strategy, HSW employee engagement, HSW strategy efficacy, leadership, and the conceptual framework.

The transcribed data produced a cell of text that was then analyzed and explored in order to gain insights, understand each of the variables and their operational definition, and identify any new areas to investigate in the quantitative phase. On the completion of the analysis, a de-identified summary report of the findings was provided to each of the participants in order to review and confirm their input was reflected in the revised conceptual HSW strategy framework.

In summary, the overall aim of the qualitative interview phase was to explore and confirm the variables, attributes, and definitions outlined in the HSW strategy conceptual framework, in addition to informing the design of the survey to be used in phase two that will validate the HSW conceptual framework. Having developed the initial framework, the study progressed to testing and refining the revised framework (Figure 2) through an online cross-sectional survey.

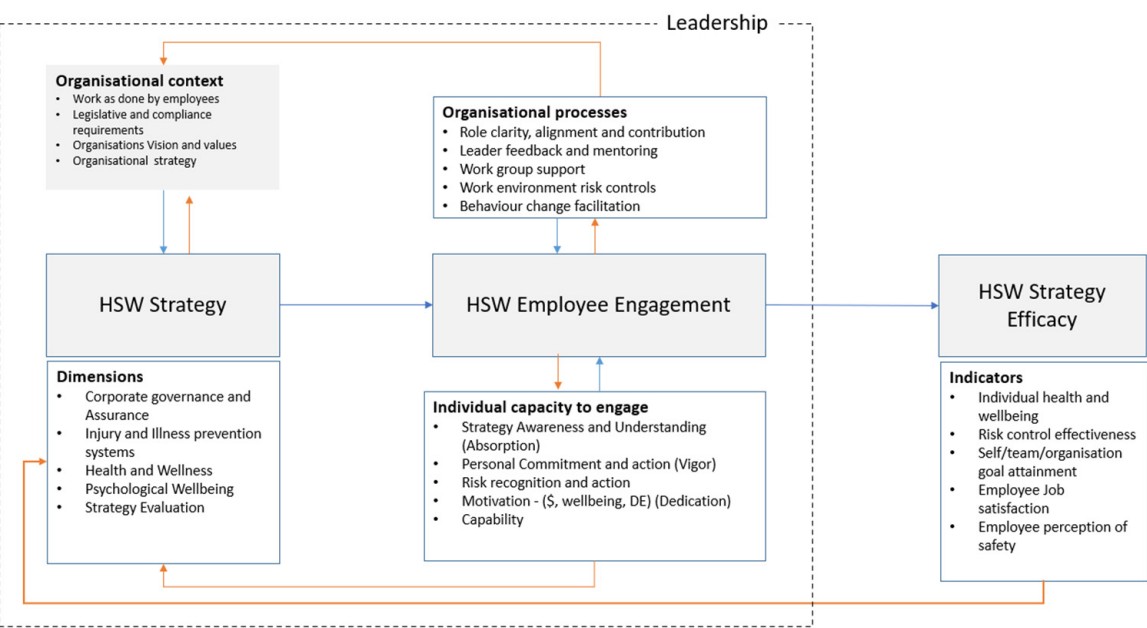

**Figure 2.** Revised HSW Strategy Framework for Phase 2.

### 3.2. Triangulation and Validation of Phase One Results (Phase Two—Quantitative Enquiry)

Phase two of the mixed method study was a cross-sectional survey used to (a) triangulate and (b) validate the qualitative findings from phase one. It sought to refine the framework from an alternative source of data, while also validating the HSW strategy and employee engagement constructs embedded in the initial framework. The second phase of the study also examined the associations between the variables presented in the initial framework through surveys, descriptive statistical analyses, and exploratory factor analyses.

The HSW Strategy and Employee Engagement Survey (WEES) was designed specifically for this study and presented as: Part A, containing seven questions relating to demographic information and professional experience; Part B (refer Appendix B), containing twenty-seven questions relating to organizational context and HSW strategy that required a response on a Likert scale ranging from (1) Strongly disagree to (5) Strongly Agree; Part C, containing seventeen questions relating to employee engagement from the Utrecht Work Engagement Scale ( UWES ) that required a response on a Likert scale ranging from (1) Strongly Disagree to (5) Strongly Agree; and Part D, containing eight questions relating to HSW strategy efficacy that required a response on a Likert scale ranging from (1) Strongly Disagree to (5) Strongly Agree.

#### 3.2.1. Participant Selection

A non-probability sampling approach was deemed as the most appropriate for the pilot and the main phase of the survey when it 'will produce findings that can be transferable to other studies' [62] (p. 15) and purposefully selects participants that are familiar with the subject matter. The criteria established to select appropriate participants and delimit the sample group was based on the following: (1) experience at a management level (middle or executive), where they were responsible for the workplace health and safety or wellbeing strategy; or (2) experience at a management level (middle or executive), where they were responsible for employee engagement or human capital management; and (3) currently working in either public or private sector organizations.

#### 3.2.2. Survey Development and Administration

The 17 themes from phase one were used to inform the development of the 27 survey questions for HSW strategy (Part B of the Survey). Before administration, the draft WEES

instrument was provided to the semi-structured interview (Phase one) participants to review the format and questions proposed for each construct. This was deemed necessary as (a) most of the questions contained in the survey had not been applied in a research setting, and (b) a similar conceptual framework to what was developed had not been tested in previous studies. Example questions for Part B derived from the qualitative phase themes were 'To what extent do you agree or disagree that Prevention of harm, including physical safety is an inherent core of worker wellbeing?' (Theme 1) and 'To what extent do you agree or disagree that Ownership enhances personal growth and the capability to engage in HSW strategy?' (Theme 13).

Pilot Survey

Piloting was a crucial step in the research process, as it provided insight into problems associated with 'wording, order, and presentation of questions that might cause respondents to provide inaccurate responses' [63] (p. 738). The survey pilot adopted a convenience sampling strategy. Initially, seven (7) respondents from phase one completed the pilot survey. Snowball sampling was then utilized to increase the responses by requesting the initial group to provide the survey link to known cases that met the sample group criteria for completion. The pilot phase served to modify the main survey instrument and ensure that it adequately captured the associations between the variables and understanding the who, what, and why of the population [64]. Consequently, the pilot phase was able to confirm that the instrument was suitable for the administration of the main survey.

Main Survey

The main survey was conducted via an anonymous online platform (Lime Survey software, version 4, LimeSurvey GmbH, Hamburg, Germany). Participants for the main survey were recruited through purposive sampling. The survey was provided to the phase one participants and promoted through LinkedIn, the Australian Institute of Human Resources discussion board on LinkedIn, through the Australian Institute of Managers and Leaders, and the Australian Institute of Health and Safety. As noted in the literature [65], the recruitment of participants that have specialized knowledge and/or leadership in their fields typically yield low response rates. Despite this challenge, the study was able to recruit 132 responses and accept 95 complete responses (72%).

Data Analysis

On the completion of the pilot and main survey, the data were checked to confirm the sampling criteria was met and the completeness of each case. The cases that did not meet the criteria or at least 95% completion of the survey was discarded (n = 37). The remaining data were then cleaned and screened using Statistical Package for Social Sciences (SPSS) software (version 26, IBM, New York, NY, USA). The data were checked for missing values and the normality of distribution using descriptive statistics, including skewness and kurtosis values and probability plots. No further cases were deleted.

An exploratory factor analysis (EFA) was conducted using the Maximum Likelihood method, due to the normality of the dataset [66], and because it allowed for the computation of a wide range of indexes of the goodness of fit, statistical significance testing of factor loadings and correlations among factors, and the computation of confidence intervals [67]. Oblique (Oblimim) rotation was applied to achieve the most parsimonious model because the items were deemed to be mostly correlational in nature when rotated [66,68].

The 27 questions measuring HSW strategy were analyzed, the rotated solution producing a seven-factor model with eigenvalues of greater than one, explaining 52 percent of the variance. The Kaiser–Meyer–Olkin Measure of Sampling Adequacy was good (0.70), and Bartlett's Test of Sphericity ($x^2$ = 949.114, df = 351, $p$ < 0.000) indicated that the data fit the model well and the appropriateness of factor analysis and extraction. The instrument yielded a high reliability statistic with a Cronbach's alpha of 0.86. Each of the factors was

reviewed based on the current theory to determine the construct they best represented based on the structure matrix loadings.

## 4. Results

### 4.1. Sample Demographics

Of the 95 valid responses, 69.5% were male, with the majority being senior managers (48.5%) with 10 years or more experience on the job (82.1%). This suggested they were well advanced in their career at a senior decision-making level, and were suitably positioned to respond to the questions on HSW strategy development and employee engagement in organizations. Most (79%) had postgraduate qualifications, suggesting advanced theoretical understanding of certain key concepts.

The majority of the 95 responses were from Queensland (43.2%), with 49.5% of responses in the 'Other' industry type, which included "high risk" industries such as utilities, power transmission, and resources/energy organizations.

Table 2 provides the demographics frequency breakdown for position, discipline, gender, location, industry type, experience, and education.

**Table 2.** Demographics position, discipline, gender, location, industry type, experience, and education.

| Position | Frequency | % of Total |
|---|---|---|
| Senior Manager | 46 | 48.5 |
| Practitioner | 30 | 31.5 |
| Manager | 19 | 20.0 |
| **Discipline** | **Frequency** | **% of total** |
| Workplace Health and Safety | 44 | 46.4 |
| Other | 31 | 31.6 |
| Human Resources | 18 | 18.9 |
| Wellbeing or Health | 2 | 2.1 |
| **Gender** | **Frequency** | **% of total** |
| Male | 66 | 69.5 |
| Female | 29 | 30.5 |
| **Location** | **Frequency** | **% of total** |
| Queensland | 41 | 43.2 |
| Victoria | 18 | 18.9 |
| New South Wales | 17 | 17.9 |
| South Australia | 6 | 6.3 |
| Western Australia | 4 | 4.2 |
| Tasmania | 4 | 4.2 |
| Northern Territory | 3 | 3.2 |
| Australian Capital Territory | 2 | 2.1 |
| **Industry Type** | **Frequency** | **% of total** |
| Other | 47 | 49.5 |
| Public | 23 | 24.2 |
| Manufacturing | 7 | 7.4 |
| Resources | 7 | 7.4 |
| Construction | 6 | 6.3 |
| Transport | 5 | 5.3 |
| **Experience** | **Frequency** | **% of total** |
| 10 or more | 78 | 82.1 |
| 5–9 | 10 | 10.5 |
| 0–4 | 7 | 7.4 |
| **Education level** | **Frequency** | **% of total** |
| Postgraduate | 75 | 79.0 |
| Undergraduate | 13 | 13.6 |
| Vocational | 7 | 7.4 |

Source: Developed for this research

*4.2. HSW Strategy Scale*

For the 27 items measuring HSW strategy, 19 (70%) reported a mean score of four or above (Agree or Strongly Agree). The question that the sample group mostly agreed with was 'Leadership influences Organizational context, work health, safety, and wellbeing strategy and employee engagement' (M = 4.59, SD = 0.78). This result was consistent with the views of most participants from the semi-structured interview phase and the literature review, in that leadership effects strategy understanding, prosocial safety behavior, discretionary effort, wellbeing, and employee engagement. The second most supported question was 'Prevention of harm, including physical safety is an inherent core of worker wellbeing' (M = 4.57, SD = 0.66). In contrast, the question that respondents were least likely to agree with was 'Individual enablers influence work health, safety, and wellbeing strategy' (M = 3.73, SD = 0.92).

The examination of the means and standard deviations for the questions relating to each construct included in the HSW strategy items indicated consistent mean scores across each of the attributes, demonstrating that the instrument provided meaningful information about the attributes being studied [64]. The seven factors revealed for HSW strategy mapped well against the revised conceptual framework from phase one, supporting the proposed relationships between organizational context, strategy content, employee engagement, and strategy efficacy.

Factor one was explained by two variables with loadings of 0.976 and 0.619. This represented the physical safety and personal growth elements of HSW. This was labelled as Worker Wellbeing. Factor two was explained by one variable with a loading of −0.968 and represented the UWES (vigor, dedication, and absorption), risk recognition, proactive action, and individual capability. This was labelled as Individual Capacity. Factor three was explained by two variables with loadings of 0.795 and 0.705. This factor represented the relationships between engagement, efficacy, and strategy feedback loops in the conceptual framework. This was labelled as Engagement and Efficacy.

Factor four was explained by eight variables with loadings ranging between −0.408 to −0.797, representing the organizational context in which the HSW strategy emerges, and the content dimensions for safety, wellness, and wellbeing from the conceptual framework. This was labelled as Strategy Context and Content. Factor five was explained by five variables with loadings ranging between 0.441 to 0.685. This represented attributes such as leader-member coaching, mentoring, strength of interpersonal relationships, and the individual's ability to influence and act on HSW. This was labelled as Connection and Ownership.

Factor six was explained by two variables with loadings of −0.638 and −0.599. This factor represented the relationships between engagement and efficacy and organizational processes. This was labelled as Engagement and Processes. Factor seven was explained by two variables with loadings of 0.537 and 0.527. This factor represented processes inclusive of governance, resourcing requirements, and the involvement of employees at various levels in the organization in strategy to determine the content. This was labelled as Strategy Process Content.

In summary, the seven factors revealed for HSW strategy were consistent with the framework derived from the literature and from phase one of the study. An examination of the means and standard deviations for the questions relating to each construct (Organizational context, HSW strategy, etc.) indicated consistent mean scores across each of the attributes, indicating that the instrument provided meaningful information about the attributes being studied [69], and confirming the frameworks internal validity.

Table 3 outlines the mean, standard deviation, and factor item loadings for the Structure matrix across each of the seven factors.

**Table 3.** Rotated solution mean, standard deviation, and factor item loadings.

| Question | Mean (M) | Standard Deviation (SD) | Factor | | | | | | |
|---|---|---|---|---|---|---|---|---|---|
| | | | 1 | 2 | 3 | 4 | 5 | 6 | 7 |
| Individual risk awareness and proactive action are central to personal growth in work health, safety, and wellbeing capability | 4.28 | 0.76 | 0.976 | | | | | | |
| Prevention of harm, including physical safety is an inherent core of worker wellbeing | 4.57 | 0.66 | 0.619 | | | | | | |
| Individual enablers influence work health, safety, and wellbeing employee engagement | 4.17 | 0.74 | | −0.968 | | | | | |
| Individual enablers influence work health, safety, and wellbeing strategy | 3.73 | 0.92 | | | 0.795 | | | | |
| Work health, safety, and wellbeing strategy efficacy influences work health, safety, and wellbeing strategy | 3.80 | 0.77 | | | 0.705 | | | | |
| Leadership influences organizational context, work health, safety, and well-being strategy and employee engagement | 4.59 | 0.78 | | | | −0.797 | | | |
| Work health, safety, and wellbeing strategy influences work health, safety, and wellbeing employee engagement | 4.27 | 0.73 | | | | −0.701 | | | |
| Organizational culture influences work health, safetym and well-being strategy development over the short and long-term | 4.43 | 0.68 | | | | −0.623 | | | |
| Individual leadership capability affects wellbeing and the level of engagement in strategy | 4.54 | 0.70 | | | | −0.582 | | | |
| Organizational processes influence work health, safety, and well-being employee engagement | 4.22 | 0.80 | | | | −0.546 | | | |
| To be in an optimal state of wellbeing, employees need to be connected at the individual, team and organizational levels and have purpose in their work | 4.48 | 0.68 | | | | −0.542 | | | |
| Organizational context influences work health, safety, and well-being strategy | 4.38 | 0.59 | | | | −0.419 | | | |
| Worker wellbeing includes employees managing lifestyle health and psychological health risks as an organizational priority, which positively affects employee commitment | 4.46 | 0.73 | | | | −0.408 | | | |
| Meaningful consultation for understanding the work health, safety, and well-being strategy implementation impacts on the level of employee engagement in the short and long term | 4.30 | 0.63 | | | | | 0.685 | | |
| Work health, safety, and well-being strategy measurement must focus on broader outcomes related to individual wellbeing, work completed, worker perceptions on safety systems, and risk management effectiveness and safety | 4.18 | 0.73 | | | | | 0.667 | | |
| Organizational and leader trust is dependent on values-based feedback which affects employee motivation and individual wellbeing. | 4.29 | 0.68 | | | | | 0.535 | | |
| Ownership enhances personal growth and the capability to engage in work health, safety, and well-being strategy | 4.35 | 0.61 | | | | | 0.441 | | |
| Personal risk awareness and control needs to be facilitated by the organization as part of strategy implementation to engage employees in work health, safety, and well-being | 4.26 | 0.62 | | | | | 0.416 | | |
| Work health, safety and well-being employee engagement influences organizational processes | 3.87 | 0.85 | | | | | | −0.638 | |
| Work health, safety, and well-being employee engagement influences work health, safety, and well-being strategy efficacy | 4.36 | 0.76 | | | | | | −0.599 | |

**Table 3.** *Cont.*

| Question | Mean (M) | Standard Deviation (SD) | Factor | | | | | | |
|---|---|---|---|---|---|---|---|---|---|
| | | | 1 | 2 | 3 | 4 | 5 | 6 | 7 |
| Work health, safety, and well-being strategy and resource allocation must be integrated and address immediate risks prior to longer term strategic risks. | 4.00 | 0.88 | | | | | | | 0.537 |
| Employees need to be involved in work health, safety, and well-being strategy development at an early stage and be clear on their personal contribution as it relates to vision, mission, and goals | 4.15 | 0.87 | | | | | | | 0.527 |

Source: SPSS version 26 output Structure matrix

## 5. Discussion

As outlined, the relationship between organizational and HSW strategy is complex, with a paucity of previous research into this area. The present study attempted to develop a strategy-centric framework that placed employee health, safety, and wellbeing as an organizational strategic priority within the broader context of the organizations' overarching strategy. The following is a discussion on the results and study limitations.

### 5.1. HSW Framework

The literature review established that a limitation of existing safety, health, and wellbeing models and frameworks was that they have largely remained detached from broader business strategy, without having a demonstrable impact beyond their primary purpose (e.g., reducing injuries). More recently, safety risk management has evolved through alternate views, such as resilience engineering, which appear to be more business-centric and provide opportunities to connect HSW with business performance. To bridge this gap, the framework for the present study was able to integrate dimensions relating to safety, health, and psychological wellbeing with those from the business area, such as resilience engineering, corporate governance, and risk management within organizational strategy.

Previous studies relating to HSW strategy have been outlined by Zou and Sunindijo [54], Yorio, Willmer and Moore [38], and the World Health Organization and Burton [55], with a focus on health and wellbeing. More recent discussion by Provan et al. [14] introduced the concept of "guided adaptability", with phases relating to anticipation, readiness to respond, synchronization, and proactive learning [14]. This approach aligns with the broader organizational resilience phases of anticipation, coping, and adaptation. Accordingly, these aspects informed the thinking on framework design because the attributes were analogous with resource-based views and enabled the creation, revision, and refinement of risk models in the strategy development to meet operational demands.

For the present study, the readiness and respond phase, outlined by Provan et al. [14], aligned with the need to match resources and capabilities to achieve organizational objectives, as outlined by RBV of strategy, whilst the synchronization and proactive learning phases were related to the feedback loops (organizational processes and individual capacity) as iterative cycles and the strategy efficacy indicators included in the framework (Factor 3). The HSW strategy process and content dimensions of the framework were related to the anticipation and readiness to respond phases. And these were found to be interrelated, supporting the need to match internal capabilities with the risk and opportunity management efforts of the organization to achieve a future state of HSW, acknowledged as an organizational strategic priority (Factor 7). Furthermore, the dynamic iterative strategy cycle adopted in the framework was reflected as organizational context (Factor 4), which corresponds with strategy development (Context and Content), outlined by Zou and Sunindijo [54], and with the anticipation, plan, and revise phase of resilience engineering [14].

The HSW, business strategy, and the employee engagement literature discussed the significant role aligned, motivated, healthy, and engaged employees play in achieving per-

formance objectives and goals and in responding to disruption (Factor 6). RBV considered that employees are a source of competitive advantage and are central to the execution of the organization's strategy and achievement of strategic goals, as demonstrated by the RBV of strategy formulation [48]. As such, the present study confirmed the employees' need to be involved in HSW strategy development (in addition to implementation) at an early stage, because they are best placed to provide insights into the nature of work, risks, and opportunities and to understand HSW requirements and implications. This furthers the established thinking around "work as imagined versus work as done" considerations outlined in "Safety 2" [25], and that employees are essential in balancing safety and organizational performance and responding to disruption (Factor 5). It also emphasizes that early involvement provides employees with clarity on their contribution to achieving the organization's strategic objectives, in addition to HSW, thereby supporting the fact that the RBV approach to strategy formulation can be applied to HSW. By doing so, it ensures that employees are clear on their contribution to achieving objectives and goals through a clear "line of sight", which is crucial for strategy implementation due to employees being able "internalize" the requirements at the individual level (Factor 2). This supports an employee's capacity to engage and their wellbeing (Factor 1).

Overall, the seven factors revealed for HSW strategy mapped well against the revised conceptual framework, supporting the proposed relationships between organizational context, strategy content, employee engagement, and strategy efficacy. In doing so, the present study was able to cohesively operationalize the zero-accident vision goal variables health and safety management system, leadership, safety culture, training, communication, and legislation [70].

In summary, the nature of work has changed, and workplace health, safety, wellbeing, and human capital practices need to ensure both individual and organizational health, manage strategic risks, and respond to disruptions in order to achieve the desired business outcomes. Therefore, a shift is required through new models and frameworks that connect HSW with business. This change also necessarily recognizes that employees are crucial to business success rather than being a risk to overcome, especially in strategy execution, which enhances their wellbeing and personal growth.

*5.2. Study Limitations*

The survey instrument utilized in phase two to test the framework presents limitations in that (a) the questions contained in the survey had not been applied in a research setting, and (b) a similar framework had not been tested, such as the one developed for this study. Because of these limitations, it was not possible to assess the Convergent and Criterion validity in this present study, which is considered desirable when developing new instruments. This limitation could be addressed in future studies by administering the survey more than once to the sample group on a longitudinal basis and comparing the results as required by the test–retest methodology. The results could be compared with the findings of this study to further develop the internal and external validity of the framework, and to explain the relationships within the framework.

Additionally, this present study was not able to achieve the target sample size in phase two. This was due to the difficulty in the recruitment of participants, which meant that a complete sample population was not able to be determined or accessed, despite utilizing several methods to increase the sample size. This is a noted challenge in organizational and business research, and this affects the degree of the homogeneity of the sample and the subsequent representativeness of the population. This also reduces the viability of using exploratory factor analysis in theory development and validating the survey instrument used. Such limitations could be addressed in the future through discipline-specific research by recruiting participants on a longitudinal basis by partnering with an employer, employee, or research bodies to access a more complete sample frame and meet sampling quotas.

## 6. Conclusions

This study sought to address the gap in the literature related to conceptualizing and deriving the definition and operationalization of HSW strategy within the organizational strategy context. Importantly, the study was able to address the calls for further research into HSW from a business perspective, because of a paucity of research with a limited focus on health and wellbeing as a priority within organizational strategy. The seven-factor framework contributes to safety, health, and wellbeing, business strategy, and the human capital literature by integrating several theories, models, and frameworks. More specifically, this study provided new professional practice insights by demonstrating HSW and its relationship with organizational strategies and employee engagement. This resulted in an industry-confirmed indicative framework (Figure 2) that provides a business-centric roadmap for organizations to assess their current approaches to HSW strategy, how employees are engaged in the process, and linkages with individual and business outcomes.

From a practice perspective, the final framework was not able to be applied within situated research settings. Further research is necessary to examine the relationships between the components, support refinement of the framework, and evaluate its impact on organizational improvement. Research opportunities may include the following: (i) conducting a broad interdisciplinary study across a cross section of industries in order to refine the framework; (ii) extending the measurement component of the framework to evaluate the impact and efficacy of the framework; and (iii) applying the framework and evaluating the short- and long-term benefits of the framework in a variety of professional settings.

**Author Contributions:** Conceptualization, B.H. and L.v.d.L.; methodology, B.H., L.v.d.L. and A.R.; formal analysis, B.H. and L.v.d.L.; investigation B.H., L.v.d.L. and A.R.; resources, B.H.; data curation, B.H. and L.v.d.L.; writing—original draft preparation, B.H. and L.v.d.L.; writing—review and editing, B.H., L.v.d.L. and A.R.; supervision, L.v.d.L. and A.R.; project administration, B.H.; funding acquisition, L.v.d.L. All authors have read and agreed to the published version of the manuscript.

**Funding:** This research was funded through Australian Government Research Training Scheme.

**Institutional Review Board Statement:** The study was conducted in accordance with the Declaration of Helsinki and approved by the Ethics Committee of the University of Southern Queensland (H18REA120, 24/05/2018) for studies involving humans.

**Informed Consent Statement:** Informed consent was obtained from all subjects involved in the study.

**Data Availability Statement:** The data presented in this study are openly available from the University of Southern Queensland Exemplar Repository at https://doi.org/10.26192/fmfm-5e77 (accessed on 20 December 2023).

**Conflicts of Interest:** The authors declare no conflict of interest. The funders had no role in the design of the study; in the collection, analyses, or interpretation of data; in the writing of the manuscript; or in the decision to publish the results.

## Appendix A. Semi Structured Interview Questions

1. In your experience, does the definition of organizational context represent the topic for this research?
2. In your experience, does the definition of Worker Wellbeing represent the topic for this research?
3. In your experience, does the definition of HSW engagement represent the topic for this research?
4. In your experience, does the definition of HSW strategy represent the topic for this research?
5. In your experience, does the definition of HSW efficacy represent the topic for this research?
6. Based on your views for Q1-Q5, does the framework outlined represent a strategic approach to HSW beyond traditional management practices?

## Appendix B

**Table A1.** HSW strategy framework survey questions (Part B).

| Question | Strongly Disagree | Disagree | Neutral | Agree | Strongly Agree |
|---|---|---|---|---|---|
| Organizational Context influences Work Health, Safety, and Wellbeing Strategy | | | | | |
| Work Health, Safety, and Wellbeing Strategy influences Work Health, Safety, and Wellbeing Employee Engagement | | | | | |
| Work Health, Safety, and Wellbeing Employee Engagement Influences Work Health, Safety, and Wellbeing strategy efficacy | | | | | |
| Organizational processes influences Work Health, Safety, and Wellbeing Employee Engagement | | | | | |
| Individual Enablers influences Work Health, Safety, and Wellbeing Employee Engagement | | | | | |
| Work Health, Safety, and Wellbeing Employee Engagement influences Organizational Processes | | | | | |
| Work Health, Safety, and Wellbeing Employee Engagement influences Individual Enablers | | | | | |
| Organizational Processes influences Organizational Context | | | | | |
| Individual Enablers influence Work Health, Safety, and Wellbeing Strategy | | | | | |
| Work Health, Safety, and Wellbeing Strategy Efficacy influences Work Health, Safety, and Wellbeing Strategy | | | | | |
| Leadership influences Organizational Context, Work Health, Safety, and Well-being Strategy and Employee Engagement | | | | | |
| Prevention of harm, including physical safety is an inherent core of Worker Wellbeing. | | | | | |
| Worker Wellbeing includes employees managing lifestyle health and psychological health risks as an organizational priority. Which positively affects employee commitment | | | | | |
| To be in an optimal state of wellbeing, employees need to be connected at the individual, team and organizational levels and have purpose in their work | | | | | |
| Individual risk awareness and proactive action are central to personal growth in HSW Capability | | | | | |
| Organizational Culture influences HSW strategy development over the short and long-term | | | | | |
| Organizational Context is dynamic and affects the short and long-term HSW strategy content. | | | | | |
| Employees need to be involved in HSW strategy development at an early stage and need to be clear on their personal contribution as it relates to vision, mission, and goals | | | | | |
| Legal obligations and organizational corporate governance requirements need to be understood and assessed as they influence the focus of strategy | | | | | |
| Individual leadership capability affects wellbeing and the level of engagement in strategy | | | | | |
| HSW strategy and resource allocation must be integrated and address immediate risks prior to longer term strategic risks. | | | | | |
| Line management drive and affect strategy implementation by translating and communicating organizational requirements for individuals and teams | | | | | |
| Ownership enhances personal growth and the capability to engage in HSW Strategy | | | | | |
| Personal risk awareness and control needs to be facilitated by the organization as part of strategy implementation to engage employees in HSW | | | | | |

**Table A1.** *Cont.*

| Question | Strongly Disagree | Disagree | Neutral | Agree | Strongly Agree |
|---|---|---|---|---|---|
| Meaningful consultation for understanding the HSW Strategy implementation impacts on the level of employee engagement in the short and long term | | | | | |
| Organizational and leader trust is dependent on values-based feedback which affects employee motivation and individual wellbeing. | | | | | |
| HSW Strategy measurement must focus on broader outcomes related to individual wellbeing, work completed, worker perceptions on safety systems, risk management effectiveness, and satisfaction with work | | | | | |

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
