# Peer review of "Prioritizing Work Health, Safety, and Wellbeing in Corporate Strategies: An Indicative Framework"

_safety_

Round 1
Reviewer 1 Report
Comments and Suggestions for Authors
Please see the attached file.

Author Response
Reviewer 1 Comment |
Author Response |
Grammatical corrections and editing |
All corrections in text have been made as per highlights by the reviewer |
The abstract should contain enough detail to understand the work that was carried out. Please consider revising. |
Agreed. Abstract has been amended to reflect the comment. |
...or accidents? It is important not to mix both concepts as incidents' management do not have a legal framework (yet) |
Actioned to refer to accidents in the manuscript. |
Organizations vs Organisations |
· Noted. Unified to American English |
Literature review - Any particular research method? |
· Noted. This section has been re-written to reflect the literature review process. |
Avoid using 1st person in scientific writing |
· Actioned. Sentences throughout document have been rewritten to remove 1st person. |
Methodology would be a more suitable title perhaps |
· Actioned. Changed to Methodology |
· Why only eight? What was the inclusion criterion? From which industries? |
· Noted. Phase one section has been rewritten to clarify and address the reviewer feedback. |
· What type of assessment was conducted? |
· Noted. Phase one section has been rewritten to clarify and address the reviewer feedback. |
· Please provide the references to the instruments |
· Noted. References are included. |
Instead of listing here the characteristics, it would be interesting to first describe the instruments, their application, collected data and data treatment and then participant selection as readers would have a general idea of the method itself |
· Noted. This section has been rewritten to clarify and address the reviewer feedback. |
Based on the views of Williams et al., [55], Costello and Osborne [56], and Hertzog 301 [57] 200 responses were considered as optimum, but 100 responses deemed satisfactory for the main survey in the study - this is just half. It is satisfactory to whom? |
Noted. Reference to 100 responses deemed satisfactory has been removed and included as a study limitation in the discussion section. |
My overall comment to the Methodology section is that is a bit hard to follow in terms of individual, sequential steps. Would it be possible to restructure/rearrange it so it would be easier to follow? |
· Noted. This section has been rewritten to clarify and address the reviewer feedback |
Gender- Not my field of expertise but I believe there is another, more appropriate, term for this. |
· Not Agreed. Gender remains an appropriate for use. |
What type of industry includes "Other"? Half of your respondents are classified under this |
· Noted. The preceding section in the manuscript above the table already outlines the other industry types |
It would be important to provide this type of information (specifically related to the scale) in the methodology section |
· Noted. This section has been rewritten to clarify and address the reviewer feedback |
An Exploratory Factor Analysis (EFA) of the 27 questions measuring HSW strategy were analyzed using the Maximum Likelihood method with Oblique (Oblimim) rotation. The rotated solution produced a seven-factor model with eigenvalues of greater than one 380 explaining 52 percent of the variance - This is methodological approach description. |
Noted. This has been relocated in the manuscript. |
The reader should have the information regarding the explanation of the factors before |
· Noted. This has been relocated in the manuscript |
How can you ensure that this framework is validated considering the low sample you used, in different contexts? |
· Agreed. The section has been rewritten, including establishing sample size and further research in different context as a limitation and future research need to confirm the framework. |

Reviewer 2 Report
Comments and Suggestions for Authors
The work is apparently interesting, but it is difficult to analyse it objectively since it does not have clear objectives, either in the abstract or in the main body of the work. The abstract is also written in a confusing way, with no clear reference to methodological aspects. The authors are reminded that the design of an abstract should follow the following minimum pattern and in the stated order: Background, objectives, methods, results, conclusions
Looking at what are supposed to be the objectives (lines 154-159), the work is unclear:
1- With regard to the bibliographical review, the objectives refer to a brief review, and in section 2, line 163 it says that a systematic review was conducted. Regarding the reference to a systematic review, it is suggested that the authors consult and follow the PRISMA Statement guidelines (PRISMA (prisma-statement.org) if they really want to carry out a systematic review. With regard to the Brief Review mentioned in the objectives, we suggest that the authors look for more recent references, since of the 64 presented, only 2 are less than 3 years old and only 9 are less than 5 years old. There are also no references from 2022 or 2023.
2- The second objective, "to report on the outcomes", also needs to be clarified. The way it is defined, it seems that it was the results that led to the objectives and not the objectives that led to the design and methodology of the study.
It is therefore suggested that the authors clarify the objectives and reformulate the manuscript accordingly.
Comments on the Quality of English Language
Only minor editing of English language is .
Author Response
With regard to the bibliographical review, the objectives refer to a brief review, and in section 2, line 163 it says that a systematic review was conducted. Regarding the reference to a systematic review, it is suggested that the authors consult and follow the PRISMA Statement guidelines (PRISMA (prisma-statement.org) if they really want to carry out a systematic review. With regard to the Brief Review mentioned in the objectives, we suggest that the authors look for more recent references, since of the 64 presented, only 2 are less than 3 years old and only 9 are less than 5 years old. There are also no references from 2022 or 2023. |
· Noted. This section has been re-written to reflect the literature review process. The wording “systematic” has been removed as it was not based on the PRISMA guidelines in the study origin. ·The references have been refresher to include more recent references where appropriate, this includes 2022/2023. |
2- The second objective, "to report on the outcomes", also needs to be clarified. The way it is defined, it seems that it was the results that led to the objectives and not the objectives that led to the design and methodology of the study. It is therefore suggested that the authors clarify the objectives and reformulate the manuscript accordingly. |
· Noted. This section has been rewritten and the specific objectives and research questions included. |

Reviewer 3 Report
Comments and Suggestions for Authors
For the first sentence, “The safety and well-being of an organisation’s people [talent] are key…”, you may state “organisation’s employees” or “organisation’s employees and talents...”
Could you improve the sentences, “In recognition of the limitations of traditional approaches [risk mitigation foci and narrow understanding of benefits]”, “…risk models that enable the use of resources [including talent]…”. Please do not use [] in the text, if it is avoidable.
The research gap presented in this work seemed to be weak to indicate the reasons why you need to conduct this work. There are amply of existing works related to Work Health, Safety, and Well-being strategy. You need to explicitly indicate why your work is needed.
Regarding the method of phrase 1, Literature Review, the process of this phrase needed to be clearly elucidated. For instance, what electronic databases were used for the paper selection and extraction?
Why in the “Materials and Methods” section, you stated “Eight semi-structured interviews with industry experts… 17 themes relating to each construct of the initial framework...”. This should be the content of the results. What you need to state in Materials and Methods are the type of methods and how would you run the procedure instead of providing the information of the already completed procedures.
In page 12, you presented a number of factors and their loadings. In your discussion, you did not clearly explain the results of the findings related to these factors.
Author Response
For the first sentence, “The safety and well-being of an organisation’s people [talent] are key…”, you may state “organisation’s employees” or “organisation’s employees and talents...” |
· Agreed. The section has been rewritten; however, it should be noted that human capital and business/organisational academia and practice have shifted focus to talent as distinct to employees |
Could you improve the sentences, “In recognition of the limitations of traditional approaches [risk mitigation foci and narrow understanding of benefits]”, “…risk models that enable the use of resources [including talent]…”. Please do not use [] in the text, if it is avoidable. |
Noted. This section has been rewritten. |
The research gap presented in this work seemed to be weak to indicate the reasons why you need to conduct this work. There are amply of existing works related to Work Health, Safety, and Well-being strategy. You need to explicitly indicate why your work is needed. |
This section has been rewritten to reflect the need for this work. It noted in the feedback that ‘there are amply works related to Work Health, Safety, and Well-being strategy’. The literature review and subsequent interviews suggested otherwise, and it should be noted this strategy framework presented is located with organisational strategy and is multidisciplinary in nature drawing on business and human resource practices, including a focus on employee engagement which to date has not been significantly discussed or applied in practice to HSW strategy, as such it is not just another HSW strategy. This feedback further illustrates an evidence-practice gap in that health and safety practice often has failed to draw on other disciplines to provide business centric solutions. |
Regarding the method of phrase 1, Literature Review, the process of this phrase needed to be clearly elucidated. For instance, what electronic databases were used for the paper selection and extraction? |
Noted. This section has been re-written.
|
Why in the “Materials and Methods” section, you stated “Eight semi-structured interviews with industry experts… 17 themes relating to each construct of the initial framework...”. This should be the content of the results. What you need to state in Materials and Methods are the type of methods and how would you run the procedure instead of providing the information of the already completed procedures. |
Noted. This section has been relocated. However, it should be noted that the focus of the paper is on phase 2 of the mixed method study as stated in the paper not a detailed presentation of phase one. The inclusion of the phase one interview is intended to be a brief orientation to the quantitative phase. The semi-structured interview phase and findings were substantial. To report on phase one is a specific paper. |
In page 12, you presented a number of factors and their loadings. In your discussion, you did not clearly explain the results of the findings related to these factors. |
Noted. This section has been rewritten to link the factors to the findings (refer Discussion section). |

Reviewer 4 Report
Comments and Suggestions for Authors
Abstract
Specify the methods (interviews and online survey), the number of participants and findings from the first phase.
methods
1. Add the interview guide in the appendix. Did the participants sign a consent form? Were the interviews recorded and transcribed? By what method were they analyzed? Is it through software?
2. Who conducted the interviews? Is there a prior acquaintance between the interviewees and the researchers?
3. How were the participants recruited for the survey? Where was the link distributed? Was anonymity guaranteed?
4. Did everyone work in the same organization?
5. How many filled out the survey and how many were dropped because they did not meet the inclusion criteria? What is the response rate?
6. Add detailed description of the tool, including the reliability (Cronbach's alpha).
results
1. Findings of the qualitative part are missing. Please add.
2. Add a description of the interviewees.
discussion
1. How do the findings correspond with previous studies?
2. What are the practical recommendations? What use can be made of the tool? How will it serve organizations in the future?
3. Add the limitations of the study
Author Response
Reviewer 4 Comment |
Author Response |
Abstract Specify the methods (interviews and online survey), the number of participants and findings from the first phase. |
Agreed. Abstract has been amended to reflect the comment. It is again important to reaffirm that the purpose of the paper is to report on Phase 2 of the present study. The inclusion of phase 1 is to orient the reader that this was a broader study. Given the substantial findings of phase 1 a specific academic paper is required to report on the findings in detail. |
1. Add the interview guide in the appendix. Did the participants sign a consent form? Were the interviews recorded and transcribed? By what method were they analyzed? Is it through software? 2. Who conducted the interviews? Is there a prior acquaintance between the interviewees and the researchers? 3. How were the participants recruited for the survey? Where was the link distributed? Was anonymity guaranteed? 4. Did everyone work in the same organization? 5. How many filled out the survey and how many were dropped because they did not meet the inclusion criteria? What is the response rate? 6. Add detailed description of the tool, including the reliability (Cronbach's alpha). |
Agreed. Amendments made as per the reviewers feedback. The reliability (Cronbach's alpha) is already included in the manuscript. Note the survey questions for part B have been included as an appendix as themes from phase 1 were used to inform survey instrument development. |
1. Findings of the qualitative part are missing. Please add. 2. Add a description of the interviewees.
|
Noted. Note the survey questions for part B have been included as an appendix 2 as themes from phase 1 were used to inform survey instrument development. Given the substantial findings of phase 1 a specific academic paper is required to report on the findings in detail. |
1. How do the findings correspond with previous studies? 2. What are the practical recommendations? What use can be made of the tool? How will it serve organizations in the future? 3. Add the limitations of the study |
Noted. Discussion and conclusion sections have been rewritten to address these comments by the reviewer. |

Reviewer 5 Report
Comments and Suggestions for Authors
The authors of this manuscript re-examine the company's strategies around health, safety and wellbeing. This is indeed a problem that modern company management must face, and it also shows the importance and value of this research. However, after all, this is an academic article, so I still make the following review comments based on academic requirements.
1. In the “Abstract” section, the use of abbreviations is discouraged unless the abbreviation is generally understandable to readers. Therefore, I recommend that authors eliminate the use of abbreviations in line 16. Also, in line 17, “… Strategy framework …” should be corrected to “… strategy framework …” (in lowercase).
2. Do the authors use British English or American English (for example: organization or organization) in their English expressions? Please unify them.
3. In the “Introduction” section, I offer the following review comments.
(1) In line 62 and 97, please add the subsection code.
(2) In line 89-90, the authors make the following description:
“… “high-reliability organizations” [23], resilience engineering [24], and “Safety 2” [24,25] …”
Authors should first provide basic theoretical explanations of these theories in this section and state their relevance to HSW. This way of writing is helpful for readers to understand the content of this manuscript.
4. In the “Literature Review” section, I make the following suggestions.
(1) In this section, the authors describe the content in bullet points. This style of writing is neither customary nor encouraged. I suggest that authors divide the content description into different subsections.
(2) In line 181-182, the authors make the following description:
“… the attendant organisational capabilities are considered a source of competitive advantage …”
Please include literature citations.
(3) There are two theories, including safety 2 and resilience engineering, which are the main theoretical basis of this manuscript. However, it is a pity that the authors' elaboration of these two theories is incomplete. It will be difficult for the reader to understand how these two theories relate to this manuscript. Please replenish them. In addition, in line 184-212, in addition to expressing in the form of subsection, the authors should also make a summary before the end of this subsection to feed back the results of past research and discussion into the framework thinking of HSW.
(4) In line 234-251, these paragraphs are hierarchically different from the previous paragraph. I suggest writing it in subsection form. Also, in line 221-222, is the description of “… the UK Work Well Model (2017) and US NIOSH Total Worker Health Model (2011) …” a literature citation? If so, please use the correct literature citation expression and list it in the "References" section.
5. In the “Materials and Methods” section, I make the following suggestions.
(1) In line 266, 280, 290, 304, 310, 321 and 328, please add the subsection code.
(2) In line 263, “figure 1” should be corrected to “Figure 1”. The same situation occurs at lines 273, 433 and 487.
(3) In line 296-301, authors are asked to cancel the bullet point writing style and describe it in a paragraph.
(4) In line 296, “WHS” should be corrected to “HSW”.
(5) In line 336-337, why do the authors specifically point out the problem of missing data on gender? In addition, the relationship between this paragraph and the surrounding context is weak. Please rethink this description.
6. In the “Results” section, I make the following suggestions.
(1) In line 362, the word “latent variable” is used incorrectly in this description. I suggest removing it or changing it to "observational variable".
(2) Please express all numbers with three digits after the decimal point. In addition, please delete the 0 before the decimal point in all numerical expressions.
(3) In line 382, ​​the expression "p" value should be lowercase and italicized.
(4) In line 389, “Table 1.” should be corrected to “Table 2.”.
(5) In Table 2, there is an item whose loading at factor 3 is .54 and whose loading at factor 6 is -.51. This item should be deleted and should not be classified in any factor.
(6) In line 395-419, authors are asked to cancel the bullet point writing style and describe it in a paragraph.
7. In the “Discussion” section, I make the following suggestions.
(1) Authors are asked to cancel the bullet point writing style and describe it in a paragraph.
(2) In line 437-451, authors are asked to provide in-depth discussions based on past research results.
8. In line 496-500, please add research limitations and eliminate the bullet point writing style.

Comments on the Quality of English LanguageThe use of English in this manuscript contains many idioms that do not conform to English. There are even inconsistencies in words. I suggest that authors ask native English-speaking professionals to re-examine the entire manuscript.
Author Response
Reviewer 5 Comment |
Author Response |
1. In the “Abstract” section, the use of abbreviations is discouraged unless the abbreviation is generally understandable to readers. Therefore, I recommend that authors eliminate the use of abbreviations in line 16. Also, in line 17, “… Strategy framework …” should be corrected to “… strategy framework …” (in lowercase). |
Agreed. Abstract has been amended to reflect the comment. |
2. Do the authors use British English or American English (for example: organization or organization) in their English expressions? Please unify them. |
Agreed. Unified to American English |
3. In the “Introduction” section, I offer the following review comments. (1) In line 62 and 97, please add the subsection code. (2) In line 89-90, the authors make the following description: “… “high-reliability organizations” [23], resilience engineering [24], and “Safety 2” [24,25] …” Authors should first provide basic theoretical explanations of these theories in this section and state their relevance to HSW. This way of writing is helpful for readers to understand the content of this manuscript.
|
Agreed. Subcodes added. Section expanded to provide basic explanation of theories.
|
4. In the “Literature Review” section, I make the following suggestions. (1) In this section, the authors describe the content in bullet points. This style of writing is neither customary nor encouraged. I suggest that authors divide the content description into different subsections. (2) In line 181-182, the authors make the following description: “… the attendant organisational capabilities are considered a source of competitive advantage …” Please include literature citations. (3) There are two theories, including safety 2 and resilience engineering, which are the main theoretical basis of this manuscript. However, it is a pity that the authors' elaboration of these two theories is incomplete. It will be difficult for the reader to understand how these two theories relate to this manuscript. Please replenish them. In addition, in line 184-212, in addition to expressing in the form of subsection, the authors should also make a summary before the end of this subsection to feed back the results of past research and discussion into the framework thinking of HSW. |
Agreed. Bullet Points have been removed and reformatted to paragraph style.
Reference included. Agreed. This section has been expanded within reason noting the word limits applicable to the paper. A summary has been included and discussion on the framework development.
|
In line 234-251, these paragraphs are hierarchically different from the previous paragraph. I suggest writing it in subsection form. Also, in line 221-222, is the description of “… the UK Work Well Model (2017) and US NIOSH Total Worker Health Model (2011) …” a literature citation? If so, please use the correct literature citation expression and list it in the "References" section |
Not Agreed. The format specific to work health and wellbeing is retained. It is noted that an objective of the research was to draw out the theoretical and practice gap pertaining to wellbeing in workplace safety, health and wellbeing strategy. That said, the section has been expanded and references added as requested by the reviewer. Given the depth of the present study the specific wellbeing findings are not included in detail as this would be a specific paper. |
5. In the “Materials and Methods” section, I make the following suggestions. (1) In line 266, 280, 290, 304, 310, 321 and 328, please add the subsection code. (2) In line 263, “figure 1” should be corrected to “Figure 1”. The same situation occurs at lines 273, 433 and 487. (3) In line 296-301, authors are asked to cancel the bullet point writing style and describe it in a paragraph. In line 296, “WHS” should be corrected to “HSW”. |
Agreed. All suggestions have been actioned through correction and re-writing as appropriate. |
In line 336-337, why do the authors specifically point out the problem of missing data on gender? In addition, the relationship between this paragraph and the surrounding context is weak. Please rethink this description. |
This was initially highlighted as it was the only data point missing. Gender has been removed from discussion and the preceding sections expanded. |
6. In the “Results” section, I make the following suggestions. (1) In line 362, the word “latent variable” is used incorrectly in this description. I suggest removing it or changing it to "observational variable". (2) Please express all numbers with three digits after the decimal point. In addition, please delete the 0 before the decimal point in all numerical expressions. (3) In line 382, ​​the expression "p" value should be lowercase and italicized. (4) In line 389, “Table 1.” should be corrected to “Table 2.”. (5) In Table 2, there is an item whose loading at factor 3 is .54 and whose loading at factor 6 is -.51. This item should be deleted and should not be classified in any factor. (6) In line 395-419, authors are asked to cancel the bullet point writing style and describe it in a paragraph. |
Noted. All suggestions by the reviewer have been actioned. |
7. In the “Discussion” section, I make the following suggestions. (1) Authors are asked to cancel the bullet point writing style and describe it in a paragraph. (2) In line 437-451, authors are asked to provide in-depth discussions based on past research results. |
Agreed. Bullet point style removed and discussion linking past studies to the framework results and factors has been included. Noting restrictions on word limits. |
8. In line 496-500, please add research limitations and eliminate the bullet point writing style. |
Agreed. Actioned. |

Round 2
Reviewer 1 Report
Comments and Suggestions for Authors
Nothing to add
Author Response
Manuscript has been reviewed further in efforts to improve the readability of document.
Reviewer 2 Report
Comments and Suggestions for Authors
In this second version, the presented work has resolved some of the basic limitations of the first version and introduces significant improvements. However, it still has major problems, such as the formalisation of the objectives at the beginning of the literature review chapter. As a result, it does not respond unequivocally to the objectives proposed by the authors.
a) it does not clearly and explicitly set out the operational definitions for the HSW strategy and employee involvement; b) concerning the second objective, it presents the results of the survey; c) it does not clearly and explicitly present an HSW strategy or an employee involvement framework for integration into an organisational strategy.
Concerning answering the presented research questions, the first one is responded to in the literature review. This means that this first question has already been answered by previous work, which does not justify its presentation as a research question for this work. As for the second, despite the survey and its results, the answer is left mainly to future work.
In this context, it seems that the manuscript was written to frame the survey results and not to present a research project designed from scratch in which the survey should appear to answer the objectives and posed questions.
In conclusion, I believe the presented manuscript lacks the necessary quality for publication in a journal such as Safety.
Author Response
Peer Reviewer Feedback |
Author response |
a) it does not clearly and explicitly set out the operational definitions for the HSW strategy and employee involvement; b) concerning the second objective, it presents the results of the survey; c) it does not clearly and explicitly present an HSW strategy or an employee involvement framework for integration into an organisational strategy.
|
Noted. Operational definitions developed specific to the multi-year research project were included in Table 1 as they relate to the framework. A further operational definition for Health, safety and wellbeing has been included to futher frame the study. As outlined the framework presented incorporates business, wellbeing and human capital dimensions in an attempt to provide a possible solution, it is noted in the feedback that employee involvement is referred to which is incorrect and a very different construct to employee engagement in the human capital literature.as investigated in the present study. Link to original research project outcome is as follows. https://eprints.usq.edu.au/view/uniqueauthor/26288.html
|
Concerning answering the presented research questions, the first one is responded to in the literature review. This means that this first question has already been answered by previous work, which does not justify its presentation as a research question for this work. As for the second, despite the survey and its results, the answer is left mainly to future work. |
Noted. As a mixed methods study the research questions were applicable across the research project with findings from the qual and quant phases mapping well against the framework and the final 17 findings. It is justifiable to position this research as an initial possible solution requiring futher work given the interdisciplinary approach taken. |
In this context, it seems that the manuscript was written to frame the survey results and not to present a research project designed from scratch in which the survey should appear to answer the objectives and posed questions.
|
Not agreed. the authors can confirm the study was a multiyear research project designed from scratch and therefore maintain its originality and contribution to practice and scholarly advancement. Link to original research project outcome is as follows. https://eprints.usq.edu.au/view/uniqueauthor/26288.html
|
Reviewer 3 Report
Comments and Suggestions for Authors
Accepted
Author Response
Manuscript has been reviewed further in efforts to improve the readability of document.
Research Questions relocated to further clarify their relationship with the objectives of the study.
Definitional aspects of health, safety and wellbeing and an original definition by the authors has been included to further frame the research.
Reviewer 4 Report
Comments and Suggestions for Authors
The authors addressed all the comments and made substantial revisions to the paper. It is now ready to be recommended for publication.
Author Response
Noted. Nil action
Reviewer 5 Report
Comments and Suggestions for Authors
The authors have revised the review comments I submitted last time to improve the readability and accuracy of this manuscript. However, in line 356, authors are requested to change the subsection number to "3.2".
Author Response
Noted. Correction has been made.
Round 3
Reviewer 2 Report
Comments and Suggestions for Authors
No comments
Reviewer 5 Report
Comments and Suggestions for Authors
The authors have made the necessary revisions to the manuscript based on my suggestions. I have no other suggestions.